# Pool Boiling Performance of Multilayer Micromeshes for Commercial High-Power Cooling

**DOI:** 10.3390/mi12080980

**Published:** 2021-08-18

**Authors:** Kairui Tang, Jingjing Bai, Siyu Chen, Shiwei Zhang, Jie Li, Yalong Sun, Gong Chen

**Affiliations:** Intelligent Manufacturing Engineering Laboratory of Functional Structure and Device in Guangdong, School of Mechanical and Automotive Engineering, South China University of Technology, Guangzhou 510640, China; 1213092117tom@sina.com (K.T.); 202020100156@mail.scut.edu.cn (J.B.); 201830010140@mail.scut.edu.cn (S.C.); swzhang@scut.edu.cn (S.Z.); rzlt1314@163.com (J.L.); YL.Sun1995@gmail.com (Y.S.)

**Keywords:** pool boiling, heat transfer enhancement, multilayer micromeshes, bubble nucleation

## Abstract

With the rapid development of electronics, thermal management has become one of the most crucial issues. Intense research has focused on surface modifications used to enhance heat transfer. In this study, multilayer copper micromeshes (MCMs) are developed for commercial compact electronic cooling. Boiling heat transfer performance, including critical heat flux (CHF), heat transfer coefficients (HTCs), and the onset of nucleate boiling (ONB), are investigated. The effect of micromesh layers on the boiling performance is studied, and the bubbling characteristics are analyzed. In the study, MCM-5 shows the highest critical heat flux (CHF) of 207.5 W/cm^2^ and an HTC of 16.5 W(cm^2^·K) because of its abundant micropores serving as nucleate sites, and outstanding capillary wicking capability. In addition, MCMs are compared with other surface structures in the literature and perform with high competitiveness and potential in commercial applications for high-power cooling.

## 1. Introduction

With the rapid development of microelectronic technology, the dimensions of high-tech electric devices are becoming more and more miniaturized and integrated [1,2,3,4]. This trend creates huge challenges for efficient heat dissipation in such narrow spaces. Extreme heat flux may lead to ultra-high operation temperatures, which will degrade the performance of electronics and even cause failure of the electronic system. Therefore, it is essential to develop an efficient cooling method to solve this problem. Pool boiling is an efficient and promising method to meet this cooling requirement because of its advantages in high heat dissipation capability, uniform surface temperature, and low energy consumption. Pool boiling has been widely used in industrial applications, including electronics and thermal power plants [5].

Key factors, including the heat transfer coefficient (HTC), critical heat flux (CHF), and the onset of nucleate boiling (ONB), are usually utilized to evaluate the pool boiling performance of the surface structures. The HTC, referring to the ratio between the heat flux and the wall superheat, demonstrates the heat transfer efficiency through the working fluid. When the heat flux transferred by the structures reaches the ONB, bubbles will begin to generate and depart from the surface, indicating that the liquid convective heat transfer process has transferred to a phase-change heat transfer process with a high HTC. When the heat flux continually increases to the CHF, exceeding the highest heat transfer capacity the sample can dissipate, giant bubbles will cover the entire surface of the sample and prevent the phase-change heat transfer process, which results in a violent temperature rise as the heat transfer efficiency sharply deteriorates. The phase-change heat transfer process exhibits a heat transfer efficiency one to two orders higher than traditional single-phase heat transfer. Therefore, to obtain a highly efficient boiling heat transfer, the ultimate goal is to increase the HTC and CHF and decrease the ONB of the structures.

In general, methods [6] for delaying the CHF and increasing the HTC include (a) extending surface area, (b) increasing nucleation sites density, (c) controlling wettability, and (d) enhancing capillary wicking. At present, intense research has concentrated on surface modifications to enhance the pool boiling performance [7,8,9]. Microfins and microgrooves have been developed to enhance the boiling performance by increasing the surface area. For example, Pastuszko et al. [10] fabricated a surface modification of micro-fins with foil. The experimental results indicated that the CHF of the enhanced surface increased by 130% compared with the plain sample. Sun et al. [11] developed a microgroove surface with reentrant cavities (MSRCs) by using the orthogonal ploughing-extrusion (P-E) method, which allowed the MSRCs to lower the wall superheat at the ONB and exhibited a 15.8% higher HTC than the copper plate. However, the boiling performance enhancement of such surface structures, which only extend the surface area, is limited. Recently, micro/nanostructures have also been widely developed for enhancing pool boiling experiments by controlling wettability and improving capillary performance [12,13]. Taylor et al. [14] investigated the boiling performance of superhydrophobic surfaces with micro/nanostructures. The experiment results showed that the superhydrophobic surfaces can improve the heat transfer efficiency due to the high density of nucleation sites. Shim et al. [15] fabricated two types of silicon samples with aligned silicon nanowire (A-SiNWs) and random silicon nanowire (R-SiNWs). They found that A-SiNWs can stand individually, expanding the wetted area to dissipate heat. The optimum pitch-to-diameter ratio of the microstructure was found to be 0.25, which maximized the liquid propagation velocity. Finally, the CHFs of A-SiNWs were measured to be 245.6 W/cm^2^, 178% and 26% higher than that of the plain sample and R-SiNWs, respectively. A similar study was conducted by Wen et al. [16], in which a two-level hierarchical surface (TLHS) was developed for improving bubble nucleation and liquid rewetting. TLHS features with microcavities (diameters of 1–10 µm) and micro valleys (dimension of 100 µm) exhibited an enhancement of 71% and 185% in the CHF and HTC compared to the copper plate, respectively.

Although micro/nanostructures have been proven to significantly enhance the boiling performance, they are still restricted by high costs when commercially applied to electronics. Moreover, the low durability of micro/nanostructures fails to meet the requirements of long-time cooling. The boiling performance enhancement of nanostructures begins degrading after 500 h of boiling and can be neglected after 900 h of operation [17]. Given that heat dissipation is extremely limited inside the space of light and thin electronic devices in the 5G era, a stable and cost-efficient surface modification with a high HTC and CHF in a narrow space is still very much desired. Micromeshes are highly promising candidates to meet this requirement due to their thinness, simplicity, and high heat transfer performance by increasing nucleate sites and liquid replenishment to the surface [18,19]. Zhao et al. [20] tested the boiling performance of a single layer of mesh screen. The results showed that a mesh screen can significantly enhance the boiling heat transfer performance by increase nucleate sites. Additionally, a mesh screen with 0.356 mm wire diameter and 1.23 mm wire spacing exhibited the optimum boiling performance in the study. Pastuszko [21] investigated boiling performance of the microfins covered with a layer of mesh, and optimized the aperture dimensions of the composited structures. However, few studies were systematically conducted on the boiling performance of sintered multilayer micromeshes, especially the bubbling characteristics.

Therefore, this study developed multilayer micromesh modifications with the advantages of low cost, ease of fabrication, high performance, and ultrathin thickness. The purpose of the study was to find the most advantageous multilayer micromeshes modifications for commercially cooling microelectronics and to determine the enhancement mechanism of multilayer micromeshes. Multilayer copper micromeshes (MCMs) are fabricated by sintering in a reducing atmosphere. The surface morphologies of samples were characterized by a scanning electron microscope (SEM). Pool boiling experiments were conducted, and the boiling performance of these samples was verified in comparison to a plain copper plate. The ONB, HTC, and CHF of the samples were investigated, and the visualization system analyzed the bubble generation, growth, and departure. Additionally, the optimum sample was also compared with previous studies. Finally, we found the MCMs provide outstanding boiling performance, indicating a wide potential for heat dissipation in electronic applications.

## 2. Fabrication and Experiments

### 2.1. Fabrication of MCMs

In the study, MCM samples were fabricated using a solid-state sintering method. Figure 1 shows the fabrication process, as follows: the micromeshes (mesh number of 200 and wire diameter of 30 μm), which are cost-effective and commercial products applied in many fields, were cut into pieces with the dimensions of 10 × 10 × 0.06 mm^3^; then, the micromeshes and copper plate with the dimensions of 10 × 10 × 1 mm^3^ were clamped by graphite molds. Next, the clamped samples in graphite molds were sintered in a reducing atmosphere (5% hydrogen and 95% nitrogen) at 960 °C for 2 h. During the sintering, the micromeshes and the copper plates were diffusion-bonded together. Subsequently, the samples were cleaned with acetone and DI water and dried in a vacuum oven to remove the excess water. Here, various layers of micromeshes were sintered on the copper plate for comparison, as shown in Table 1.

### 2.2. Experiments

The experimental setup was composed of a test platform, a temperature data acquisition device, a power supply system, and a high-speed camera, as shown in Figure 2. The test platform included Bakelite plates and a rectangular PC plastic vessel. The Bakelite plates were used to form the chamber and reduce heat loss within the environment. A rectangular PC plastic vessel (with the dimensions of 80 × 80 × 200 mm^3^), which provides a better perspective for observing the bubble generations by the high-speed camera, was used to store the liquid pool. K-type thermocouples (*T*1 and *T*2) on the copper block were located in the copper block to measure the temperatures, as shown in Figure 2. The dimensions of the top surface of the copper block were similar to the sample (10 × 10 mm^2^). Eight block heaters were inserted in the copper block to provide heat input power. The temperature data acquisition device was used to record the temperatures for calculations of the heat flux and heat transfer coefficient. The power supply system was responsible for adjusting the input power. It should be noted that a 200 W auxiliary heater was used in the boiling pool to assist with the control of the water temperature at the saturation condition during the experiments.

Before the test, the sample was welded on the top surface of the heating block by melting a thin layer of solder at a temperature of nearly 200 °C. The thickness of the solder layer was ensured to be 0.2 mm, which was controlled by calculating the exact mass of the solder wire. This ensured the same contact thermal resistance between the samples and the copper block in order to obtain consistent results. For the preparation of the test, water was heated in saturated boiling for at least half an hour to degas. During the test, the heat input power was applied to the block in intervals of 5 to 20 W controlled by the voltage transformer. The equilibrium state was obtained when the temperatures were recorded for 2 min with a fluctuation within 0.2 °C. The increasing heat input was continually applied in the copper block until the boiling process reached the CHF.

### 2.3. Data Reduction

According to Fourier’s Law, the heat flux *q*″ that the samples dissipate can be calculated as follows:(1)$$q^{\prime \prime }=-k_{\mathrm{cu}}\frac{dT}{dx}$$
where *k*_cu_ is the thermal conductivity of copper (393.9 W/(m·K) at 100 °C and 386.5 W/(m·K); here, we take the average value of 390.2 W/(m·K) as the final copper thermal conductivity); $\frac{dT}{dx}$ is the temperature gradient, and can be calculated using the copper block temperatures measured by two K-type thermocouples (*T*1 and *T*2):(2)$$\frac{dT}{dx}=\frac{T2-T1}{x}$$
where *x* is the distance between *T*1 and *T*2.

The wall superheat Δ*T* is the difference between the wall temperature and the surrounding water temperature at saturated points, and can be calculated as follows:Δ*T* = *T*_w_ − *T*_sat_(3)
where *T*_w_ is the wall temperature of the samples and can be obtained as follows:(4)$$T_{w}=T1-q^{\prime \prime }\left(\frac{L}{k_{\mathrm{cu}}}+\frac{\delta }{k_{s}}+\frac{\delta s}{k_{\mathrm{cu}}}\right)$$
where $k_{s}$ is the thermal conductivity of the solder (50 W/(m·K)), L is the distance from the first thermocouple *T*1 to the top surface of the heating block, δ is the thickness of the solder layer, and δs is the thickness of the sample.

The HTC (*h*) can be calculated as follows:(5)$$h=\frac{q^{\prime \prime }}{\Delta T}$$

The uncertainty can be calculated by using a standard error analysis method. K-type thermocouples have a measurement uncertainty of ±0.5 °C. The location uncertainty for those thermocouples is estimated as ±0.05 mm. By controlling the mass of the solder, the thickness uncertainty of the solder layer is within 2%. According to the operating temperature (from 100 to 250 °C), the uncertainty of the thermal conductivity of copper is less than 1%, and the uncertainty of the thermal conductivity of the solder paste is assumed to be the same with copper. According to Equations (6)–(9), at low heat flux, the uncertainties of the heat flux and HTC are estimated to be less than 20.4% and 22.1%, respectively. However, when at higher heat flux (*q*″ > 40 W/cm^2^), the uncertainties of the heat flux and HTC are estimated to be less than 4.25% and 5.94%, respectively, which correspond to the main region of interest.
(6)$$\frac{U_{q^{\prime \prime }}}{q^{\prime \prime }}=\sqrt{{\left(\frac{U_{\lambda_{\mathrm{cu}}}}{\lambda_{\mathrm{cu}}}\right)}^{2}+{\left(\frac{U_{T_{1}}}{T_{2}-T_{1}}\right)}^{2}+{\left(\frac{U_{T_{2}}}{T_{2}-T_{1}}\right)}^{2}+{\left(\frac{U_{x}}{x}\right)}^{2}}$$
(7)$$\frac{U_{T_{w}}}{T_{w}}=\sqrt{{\left(\frac{U_{T_{1}}}{T_{w}}\right)}^{2}+{\left[\left(\frac{L}{k_{\mathrm{cu}}}+\frac{\delta }{k_{s}}+\frac{\delta_{s}}{k_{\mathrm{cu}}}\right)\cdot \frac{U_{q^{\prime \prime }}}{T_{w}}\right]}^{2}+{\left(\frac{q^{\prime \prime }U_{L}}{k_{\mathrm{cu}}T_{w}}\right)}^{2}+{\left(\frac{q^{\prime \prime }(L+\delta_{s})U_{k_{\mathrm{cu}}}}{k_{\mathrm{cu}}^{2}T_{w}}\right)}^{2}+{\left(\frac{q^{\prime \prime }U_{\delta s}}{k_{\mathrm{cu}}T_{w}}\right)}^{2}+{\left(\frac{q^{\prime \prime }U_{\delta }}{k_{s}T_{w}}\right)}^{2}+{\left(\frac{q^{\prime \prime }\delta U_{k_{s}}}{k_{s}^{2}T_{w}}\right)}^{2}}$$
(8)$$\frac{U_{\Delta T}}{\Delta T}=\sqrt{{\left(\frac{U_{T_{w}}}{\Delta T}\right)}^{2}+{\left(\frac{U_{T_{sat}}}{\Delta T}\right)}^{2}}$$
(9)$$\frac{U_{h}}{h}=\sqrt{{\left(\frac{U_{q^{\prime \prime }}}{q^{\prime \prime }}\right)}^{2}+{\left(\frac{U_{\Delta T}}{\Delta T}\right)}^{2}}$$

## 3. Results and Discussion

### 3.1. Surface Characterizations of MCMs

Figure 3 shows the SEM images of the samples. It is clear that the copper plate (CP) surface is very smooth, while those sintered with copper micromeshes exhibit an amount of rectangle micropores. Figure 3b–d show the surface morphologies of MCM-1, MCM-3, and MCM-5. It is evident from these figures that the density of the micropores increases and the dimensions of single pores decrease with increasing number of micromesh layers. In addition, in the nucleation stage, those high-density microcavities enable the formation of nucleation sites and provide more liquid flow channels, which may further delay the CHF and increase the HTC.

### 3.2. Boiling Performance Enhancement

During the tests, two independent MCM-1 samples and MCM-3 samples were tested repeatedly. We observed that the boiling curves showed good consistency in saturation boiling tests using the current experimental setup. Figure 4 shows the boiling curves of the copper plate and the three multilayer micromesh samples. It is clear that samples with micromeshes exhibit better boiling performance than the copper plate. At low heat flux, the wall superheat required to evoke the ONB of the copper plate is 12.0 °C, while those of MCM-1, MCM-3, and MCM-5 are approximately 11.2, 5.4, 4.5 °C, respectively. It is evident from Figure 5 that at the ONB, several large bubbles were generated from the surface of the CP, while abundant small bubbles were generated from sintered micromesh surfaces at similar heat flux. This indicates that surfaces with micromeshes more easily induce the ONB. The main reason is that rectangle micropores formed by micromeshes provide easily activated bubble nuclei. In the study, MCM-5 required the lowest wall superheat to evoke the ONB at 4.5 °C compared to MCM-1 and MCM-3. This is attributed to the smallest micropores of MCM-5, which provide easier-activated nucleation sites and facilitate bubble nucleation at low heat flux. Another feature that can be found in Figure 4 is that the boiling curves of MCM-5 and MCM-3 exhibit turning points at the superheat of 4.5 and 5.4 °C, respectively, while the boiling curves of CP and MCM-1 present a continuous upward trend as the heat flux increases during the initial heating stage. The sudden drop in the wall superheat of MCM-3 and MCM-5, with the increase in the heat flux implies that a rapid phase-change heat transfer process appeared at such a heat flux and dissipated the amount of heat, thereby decreasing the wall superheat.

As the heat flux continually increases to the CHF, a large bubble generates and covers the whole surface of samples, as shown in Figure 5. At the CHF, the samples were isolated from the working liquid and could not dissipate the heat efficiently. Figure 3 illustrates that multilayer micromesh surfaces can reach a much higher CHF than plain copper plates and that the CHF increases as the layer number of the micromeshes increases. The CHF of MCM-5 is 207.5 W/cm^2^, 120.5% higher than that of copper plate, which can only reach 94.1 W/cm^2^. Other samples, such as MCM-1 and MCM-3, also exhibited CHF improvements of 78.3% and 89.2%, respectively. The enhanced CHF is attributed to an increased surface area and an improved wicking capillary performance, which result from micropores formed by multilayer micromeshes. Moreover, as the layer number of the micromeshes increases, the size of the micropores decreases, thus resulting in a higher capillary force of the micromesh surface. Additionally, more layers of micromeshes provide more liquid-replenishing channels, further delaying the CHF.

Figure 6 shows the HTC as a function of the heat flux on the samples. It is obvious that the multilayer micromesh samples have a higher HTC, especially in the moderate-to-high heat-flux region. For example, at a heat flux of nearly 20 W/cm^2^, MCM-3 and MCM-5 demonstrated an HTC of 3.68 W/(cm^2^·K) and 4.69 W/(cm^2^·K), respectively, which are 160% and 330% higher, respectively, than that of the copper plate (1.41 W/(cm^2^·K)). This is because at low heat flux, a rapid boiling process occurred and dominated the heat transfer for MCM-3 and MCM-5, which can also be observed in Figure 5, while only a few large bubbles generated on the surface of the CP. MCM-1 produced an increase of 30% in the HTC compared to the CP at low heat flux because bubble nucleation sites resulting from the large micropores are not completely activated at such heat flux. In the high heat flux region, particularly near the CHF, the multilayer micromesh samples presented substantially superiority to the copper plate. MCM-1, MCM-3, and MCM-5 presented a highest HTC of 8.4 W, 14.7, and 15.6 W/(cm^2^·K), respectively, while CP exhibited a highest HTC of only 4.5 W/(cm^2^·K). This is because, at high heat flux, all the nucleation sites formed by the micromeshes are active, subsequently intensifying the boiling process and efficiently dissipating the heat. Additionally, the HTC of the multilayer micromeshes increases as the layer number increases. This is mainly attributed to the decreasing size of the micropores when increasing the micromesh layers (Figure 3). Interestingly, the HTCs of MCM-5 and MCM-3 increased as the heat flux increased, but they showed a downward trend when reaching the CHF. For example, the highest HTC reached by MCM-5 was 15.6 W/(cm^2^·K) at a heat flux of 169.1 W/cm^2^, which decreased by 9.7% in the HTC of MCM-5 at the CHF. This indicates that local dry out occurs inside the porous structures, raising the surface temperature near the CHF.

### 3.3. Bubble Generation Visualization

Further studies on bubbling characteristics for MCM-1 and CP were conducted using high-speed visualization. Figure 7 shows the bubble cycles of CP and MCM-1 at low heat flux for quantitative analysis of the boiling. The bubbles are highlighted with red frames to clearly indicate the growth and departure of the bubble circulation. Figure 8 shows the variation in the bubble diameter with time on the surfaces of MCM-1 (at Δ*T* = 14.6 °C and *q*″ = 20.6 W/cm^2^) and PC (at Δ*T* = 14.6 °C and *q*″ = 18.1 W/cm^2^). It is obvious that the departure bubble on the surface of MCM-1 was much smaller than that of the CP. The diameter of MCM-1 was only 1.51 mm, approximately half of that of the CP (3.07 mm). The decreased departure diameter of the bubbles is attributed to the micropores, which serve as the small nucleus and isolate the small bubbles from the liquid. Moreover, the whole period of bubbles on the surface of MCM-1 was much shorter than that on the CP. The bubble departure frequency of MCM-1 was 115 Hz, 69.1% higher than that of CP. This is mainly because the micropores promote bubble departure by limiting the bubble’s continuous growth. In addition, the menisci evaporation is enhanced due to the increased vapor–liquid interface area. Thereby, the phase-change process on the surface of MCM-1 was more violent, and thus the heat transfer rate was higher.

### 3.4. Comparison with Other Samples

Researchers have developed various modification routines to enhance the pool boiling performance over the past decades. To identify the advantage of MCMs in this study, it was important to conduct a comparison of the MCM samples and others reported in the literature, as shown in Figure 9. Deng et al. [22] developed a re-entrant channel coating with sintered copper powder porous structures (PC-RG). The results showed that PC-RG dissipates the heat flux of 114.8 W/cm^2^ at the wall superheat of 61.8 °C. Tang et al. [23] fabricated a porous interconnected net structure, which exhibited a heat flux of 114.8 W/cm^2^ at the wall superheat of 39.15 °C. Chen et al. [24] proposed interconnected microchannels with re-entrant cavities (IMRCs) using a microfabrication method, and IMRCs transferred the heat flux of 163.6 W/cm^2^ at 21.31 °C. The aforementioned simple machined structures exhibited various enhancements in boiling performance compared to a plain copper plate. However, the boiling curve of MCM-5 is located on the upper left section, as shown in Figure 9, which means that MCM-5 has higher heat transfer efficiency compared to the other methods. In addition, MCM-5, with a smaller thickness of 0.24 mm, is more promising for heat dissipation in compact electronics. 

Wen et al. [16] manufactured a two-level hierarchical surface combining different lengths of nanowires, exhibiting a high CHF at 250 W/cm^2^ at a wall superheat of 28.5 °C. A similar CHF was also obtained using the modifications proposed by Li et al. [25]. The multiscale modifications comprising porous and nanocoated porous pillars reached the CHF at 242.1 and 252.6 W/cm^2^ at the superheat of 27.96 and 15 °C, respectively. Rishi et al. [26] utilized a combination of ball milling, salt templating, and sintering techniques to achieve a porous coating. This sample exhibited a CHF of 289 W/cm^2^ at a wall superheat of only 2.2 °C. Nanoflower structures fabricated by chemical etching presented a high CHF of 299.7 W/cm^2^ [27]. The CHF of MCM-5 is 210 W/cm^2^, lower than the micro-nanostructures proposed in Refs. [16,25], indicating that MCM samples may enable further improvement in the CHF by adding micro/nanostructures on the current mesh surfaces. Although some previous methods, particularly those concerning surface modification of nanostructures, exhibited superior boiling heat transfer performance in multilayer micromeshes and micromesh surfaces, the advantages provided by the proposed meshes (simplicity, low cost, and high durability) are still highly competitive and show considerable potential in commercial applications for cooling compact electronics.

## 4. Conclusions

This study proposed a surface sintered with multilayers micromeshes for commercially cooling compact electronics. The boiling heat transfer characteristics of the multilayer micromesh samples were investigated. The effects of layer number on boiling performance were studied, and the bubble generations were visualized. The main conclusions are summarized as follows:(1)The multilayer micromesh surfaces exhibit significant enhancement in pool boiling heat transfer, including the CHF, HTC, and ONB, compared to a plain copper plate. The micropores formed by multilayer micromeshes enhance the heat transfer performance through an enlarged surface area, increasing easily activated nucleation sites and improving capillary wicking performance.(2)Increasing the layers of micromeshes can decrease the size of the micropores, increase the density of nucleation sites, and improve capillary wicking performance, further increasing the HTC and delaying the CHF of the samples. MCM-5 exhibited the optimum boiling performance in this study, with the highest CHF of 207.5 W/cm^2^ and the highest HTC of 15.6 W/ (cm^2^·K).(3)When compared to other modifications, MCM-5, with a remarkable boiling performance (high CHF and HTC), low cost, simplicity, and high durability, shows industrial prospects for commercial compact microelectronics cooling.

## Figures and Tables

**Figure 1 micromachines-12-00980-f001:**
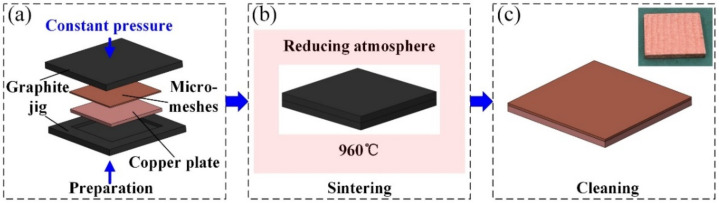
Fabrication process of the MCMs. (**a**) Preparation, (**b**) Sintering, (**c**) Cleaning.

**Figure 2 micromachines-12-00980-f002:**
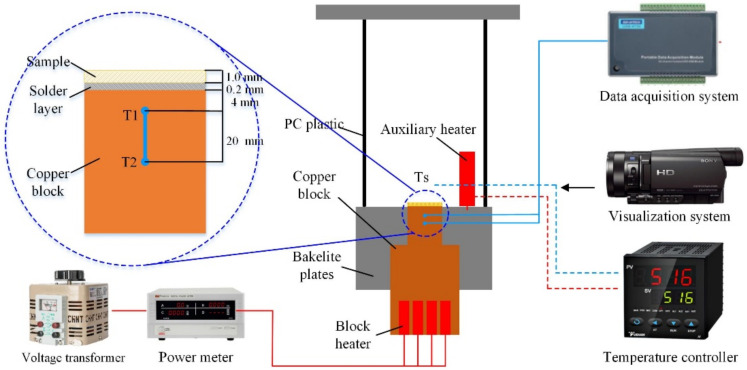
Details of the pool boiling experiment setup.

**Figure 3 micromachines-12-00980-f003:**
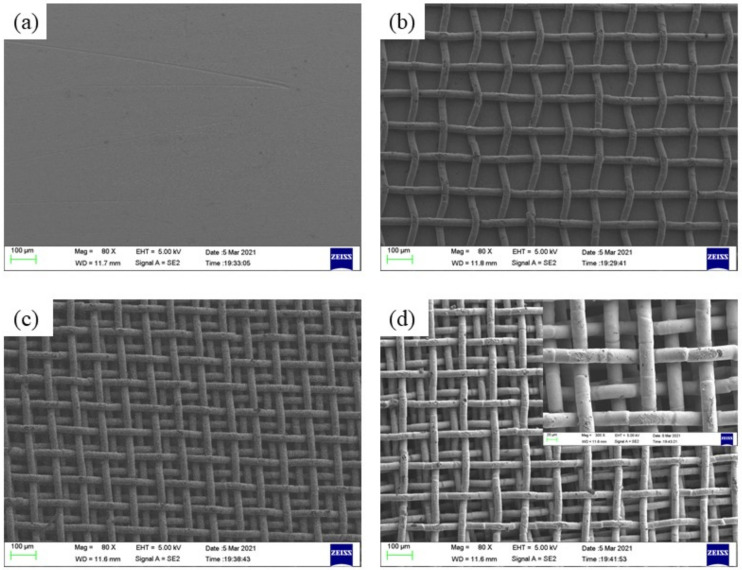
SEM images of plain and different layers meshes of samples: (**a**) CP, (**b**) MCM-1, (**c**) MCM-3, and (**d**) MCM-5.

**Figure 4 micromachines-12-00980-f004:**
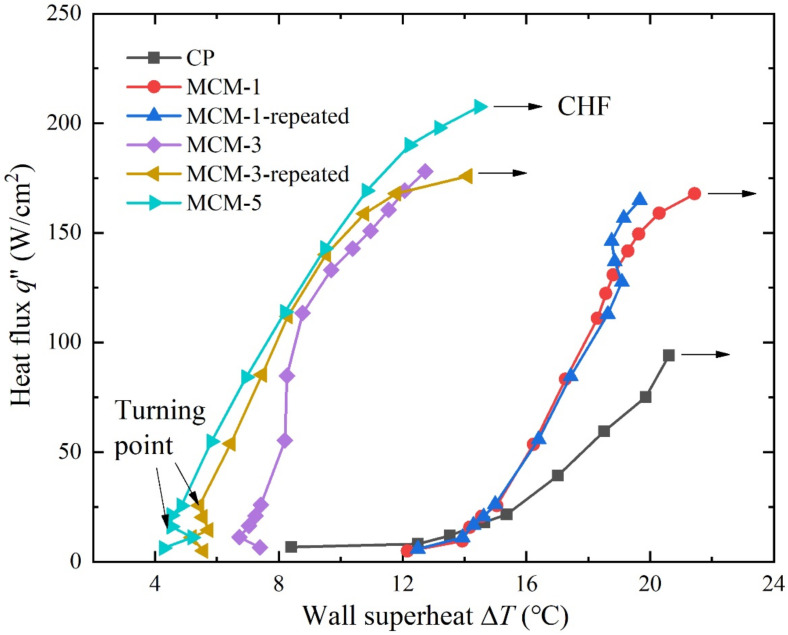
Wall superheat and heat flux relationship for CP, MCM-1, MCM-3, and MCM-5 in saturated boiling (*T*_sat_ = 100 °C).

**Figure 5 micromachines-12-00980-f005:**
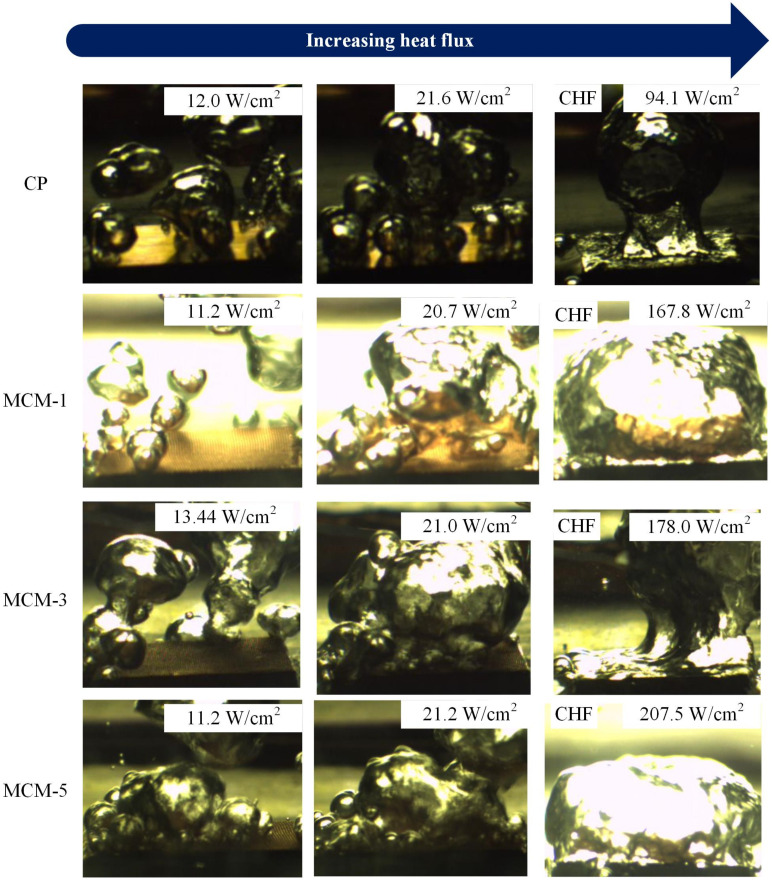
Bubbling visualizations at different heat fluxes on the CP, MCM-1, MCM-3, and MCM-5.

**Figure 6 micromachines-12-00980-f006:**
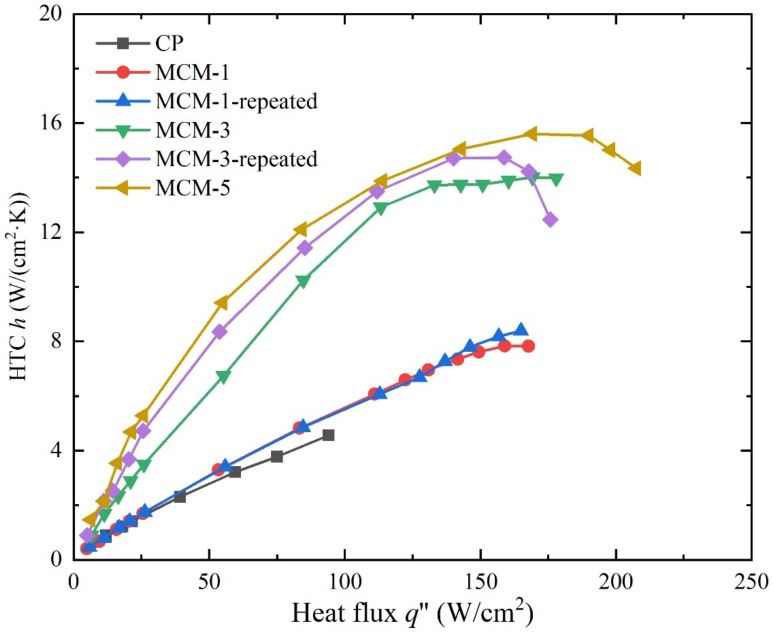
HTC and heat flux relationship for the plain sample, MCM-1, MCM-3, and MCM-5 in saturated boiling (*T*_sat_ = 100 °C).

**Figure 7 micromachines-12-00980-f007:**
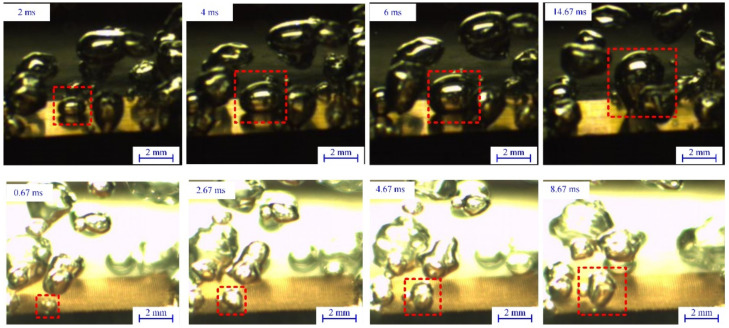
Bubble circle visualizations of the CP and MCM-1 at *q*″ =18.1 W/cm^2^ and *q*″ =20.6 W/cm^2^, respectively.

**Figure 8 micromachines-12-00980-f008:**
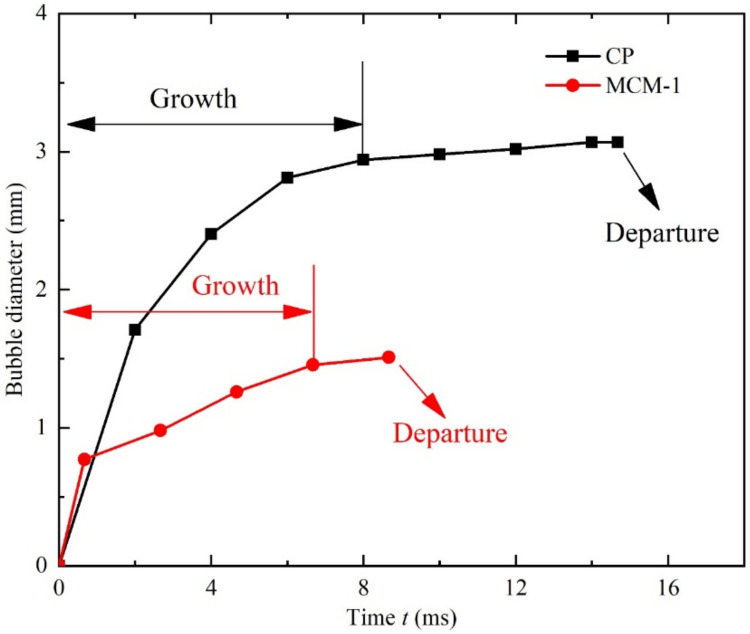
Bubble growth rate on the CP and MCM-1 surfaces.

**Figure 9 micromachines-12-00980-f009:**
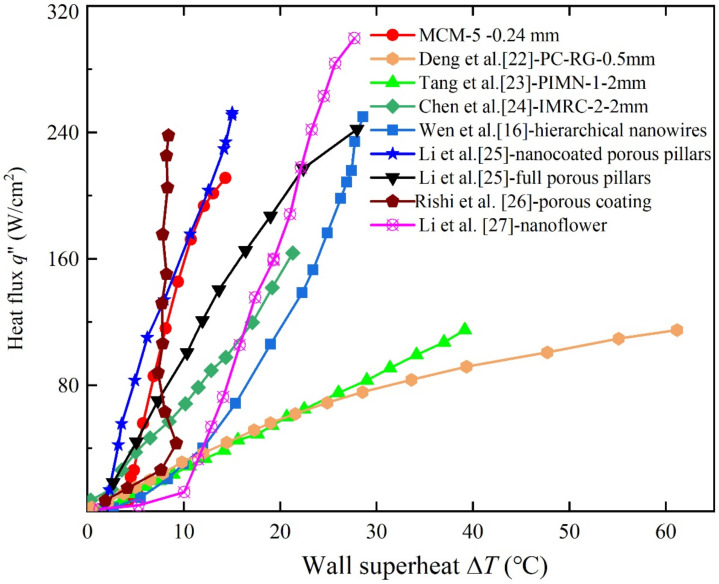
Comparison of boiling curves in the literature.

**Table 1 micromachines-12-00980-t001:** Parameters of the CP and MCMs.

Sample	Outline Dimensions (Length × Width) (mm × mm)	Layer Number of Micromesh	Thickness of Micromesh Layers (mm)
CP	10 × 10	0	0
MCM-1	10 × 10	1	0.06
MCM-3	10 × 10	3	0.16
MCM-5	10 × 10	5	0.24

## Data Availability

Data underlying the results presented in this paper are available from the corresponding authors upon reasonable request.

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
