# Peer review of "Pool Boiling Performance of Multilayer Micromeshes for Commercial High-Power Cooling"

_micromachines, 2021, doi:10.3390/mi12080980_

Round 1

Reviewer 1 Report

Please find the review in the enclosed document.

Reviewer 2 Report

  1. A complete nomenclature is needed for all abbreviations, subscripts, and parameters in the equations.
  2. The title of the manuscript should be sharper.
  3. The originality of the paper needs to be stated clearly. It is of importance to have sufficient results to justify the novelty of a high-quality journal paper. The Introduction should make a compelling case for why the study is useful along with a clear statement of its novelty or originality by providing relevant information and providing answers to basic questions such as: What is already known in the open literature? What is missing (i.e., research gaps)? What needs to be done, why and how? Clear statements of the novelty of the work should also appear briefly in the Abstract and Conclusions sections.
  4. More explanation is required in results and discussion section.

Reviewer 3 Report

  • Institutional email address should be provided instead of personal ones (163.com)
  • Professional English proofreading is mandatory, at its current state, it is hardly difficult to follow due to the poor English level.
  • Revise units within the manuscript, there are several errors
  • Avoid commercialising on Figure 2
  • Provide data range of the setup equipment
  • Authors should validate their results through extensive experimental campaign considering wider temperature range
  • Conclusions should provided further insight rather than summarising paper results

Round 2

Reviewer 1 Report

The manuscript can be accepted for publication in the present form.

Reviewer 2 Report

The manuscript is improved significantly. 

Reviewer 3 Report

Thanks for considering comments and suggestions accordingly, final English editing revision should be performed